# Lean-STaR:
# Learning to Interleave Thinking and Proving

**Haohan Lin**[2*]

**Zhiqing Sun**[1]

**Sean Welleck**[1]

**Yiming Yang**[1]

[1]Language Technologies Institute, Carnegie Mellon University
[2]Institute for Interdisciplinary Information Sciences, Tsinghua University

`{haohanl,zhiqings,yiming,swelleck}@cs.cmu.edu`

`https://leanstar.github.io/`

## Abstract

Traditional language model-based theorem proving assumes that by training on a sufficient amount of formal proof data, a model will learn to prove theorems. Our key observation is that a wealth of *informal* information that is not present in formal proofs can be useful for learning to prove theorems. For instance, humans think through steps of a proof, but this thought process is not visible in the resulting code. We present Lean-STaR, a framework for training language models to produce informal thoughts prior to each step of a proof, thereby boosting the model's theorem-proving capabilities. Lean-STaR uses retrospective ground-truth tactics to generate synthetic thoughts for training the language model. At inference time, the trained model directly generates the thoughts prior to the prediction of the tactics in each proof step. Building on the self-taught reasoner framework, we then apply expert iteration to further fine-tune the model on the correct proofs it samples and verifies using the Lean solver. Lean-STaR achieves better results on the miniF2F-test benchmark within the Lean theorem proving environment, significantly outperforming base models ($43.4\% \rightarrow 46.3\%$, Pass@64). We also analyze the impact of the augmented thoughts on various aspects of the theorem proving process, providing insights into their effectiveness.

## 1 Introduction

We introduce Lean-STaR, a framework for learning to interleave informal thoughts with steps of formal proving. Building on the Self-Taught Reasoner (STaR) framework [27], we enable language models to interleave step-by-step rationales (i.e., thoughts) [15, 23] with formal proving in a two-stage process. In an initial phase, we prompt a sufficiently capable language model, such as GPT-4 [1], and generate retrospective thoughts based on a dataset of human-written proofs, such as Mathlib, the largest collection of human-written proofs in Lean [14]. Subsequently, we fine-tune a thought-augmented tactic predictor [6, 5, 11, 9] that, given a Lean state, can generate a thought and predict the subsequent tactic. In a second phase, we optimize this thought-augmented tactic predictor with the expert iteration algorithm [2, 20], using multi-step success rate in theorem proving as the reward.

---

*Work done during the visit at CMU.

38th Conference on Neural Information Processing Systems (NeurIPS 2024).

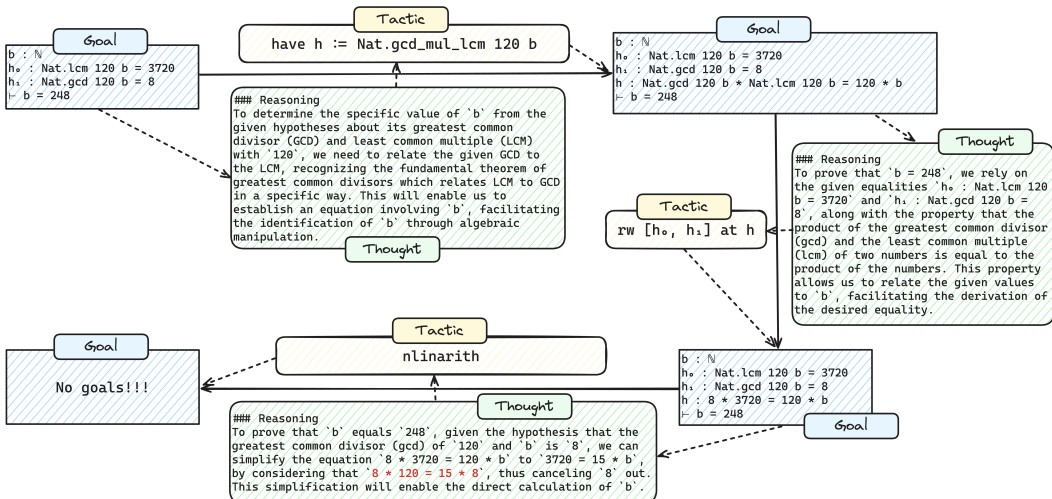

Figure 1: **An example of Lean proof and thoughts generated by Lean-STaR**. Note that there is a calculation error in the thought (in red), but this does not affect the correctness of the proof because the calculation task is actually completed by the interactive theorem prover (i.e., Lean's nlinarith) instead of the language model. This shows a benefit of combining neural and symbolic systems.

We instantiate Lean-STaR by generating roughly 50,000 thought-augmented examples from Lean's Mathlib [14], then synthesize an additional 50k examples through two iterations of expert iteration. To the best of our knowledge, this yields the first thought-augmented dataset for theorem proving. After fine-tuning an InternLM2-7b base model [26] on our thought-augmented data, our final Lean-STaR model can solve $34.8\%$ (pass@32) or $36.1\%$ (pass@64) of the problems on miniF2F-test [28]. Using stronger base model InternLM2-7b-plus, Lean-STaR can achieve $45.4\%$ (pass@32), significantly surpassing the previous results of $43.4\%$ (pass@32). In summary, Lean-STaR offers a framework for teaching language models to interleave informal thoughts with formal verification, advancing the capabilities of language models in automated theorem proving.

## 2 Our Method: Lean-STaR

We introduce Lean-STaR, a new method for combining informal thoughts with formal theorem proving.

We describe the data generation and training of the direct tactic prediction model (SFT), the thought-augmented tactic prediction model trained with synthetic data (Lean-CoT), and the final model trained with expert iteration (Lean-STaR).

### 2.1 Direct Tactic Prediction

We define the theorem-proving problem as a *Markov Decision Process* (MDP) $(\mathcal{S}, \mathcal{A}, P_a, R_a)$ where proof states serve as states in MDP and tactics serve as actions. From this perspective, a proof is a trajectory $(s_1, a_1, r_1), (s_2, a_2, r_2), \cdots$ of states $s_i$, tactics $a_i$, and rewards $r_i \in \mathbb{R}$, and the ITP (e.g., Lean) provides each new state $s_{i+1}$.

In the typical setting [18], proving a theorem consists of providing a proof state $s$ to the language model and then generating a tactic from the language model $M$, i.e., $\pi_M(a|s)$. The language model can be fine-tuned for this task using a dataset of (proof state, next-tactic) pairs from successful proof trajectories, i.e. $D = \{(s^i, a^i) : i = 1, \cdots, M\}$, where final states have a reward of 1. We refer to a language model fine-tuned on such a dataset as a *supervised fine-tuning (SFT)* model.

### 2.2 Thought-augmented Tactic Prediction

Existing approaches typically train only on formal states and tactics [18]. We hypothesize that incorporating a latent *thought* can improve a model's ability to predict the next tactic. Formally, we

introduce a hidden "thought" variable $t_i$ prior to each tactic, and then extend the model to the form $\pi_M(a_i, t_i|s_i) = \pi_M(a_i|t_i, s_i)\pi_M(t_i|s_i)$. In thought-augmented tactic prediction, the distribution over the next tactic can then be expressed as: $\pi_M(a_i|s_i) = \sum_{t_i} \pi_M(a_i|t_i, s_i)\pi_M(t_i|s_i)$.

The key challenge is obtaining (state, thought, tactic) pairs for training a model. To this end, we introduce **retrospective rationale generation**. Our motivating observation is that the distribution of natural language thoughts in theorem-proving $\pi_M(t_i|s_i)$ is scarce in the pre-training corpus of large language models. In turn, we find that even the most powerful GPT-4 model does not perform well in generating the correct rationale through few-shot prompting [7]. Given a powerful large language model $G$, which we refer to as the oracle model[2], we give the oracle model the ground-truth tactic $a_i$ and let the oracle model produce the thought $t_i$ given the current state $s_i$ and ground-truth tactic $a_i$. This helps improve the pass rate and produce thought-augmented data more efficiently. Our few-shot prompt is provided in Appendix F. The design principle of the prompt is to prevent the oracle model from generating hindsight-like thoughts. With a new retrospectively annotated dataset by the oracle model $D_T$, we obtained our first thought-augmented tactic prediction model, Lean-CoT, by fine-tuning from the SFT model.

### 2.3 Bootstrapping Thought-augmented Theorem Proving

We propose to apply expert iteration to further improve the performance of Lean-CoT. Specifically, we start from the initial Lean-CoT model $M_0$ and the initial dataset $D = \{s^i : i = 1, \cdots, M\}$, which consists of all initial states $s^i$ of the theorems to be proved. In iteration 1, we use model $M$ to sample $K$ times per problem. Each time the model will produce a proof trajectory $[(s_0, t_0, a_0), (s_1, t_1, a_1), \cdots, (s_n, t_n, a_n)]$. Then we create a new dataset $D_1$ by filtering the generated trajectories to include only the successful ones. De-duplication is then applied to the collected trajectories. Now, we can further fine-tune the SFT model $M$ on dataset $D_T \cup D_1$ to produce Lean-STaR model $M_1$. Then we can similarly produce Lean-STaR model $M_2$ from $M_1$.

## 3 Experiments

We instantiate Lean-STaR using the best available open language model pre-trained on the Lean corpus (InternLM2-Math-base-7b [26]), and follow standard practice in using Lean's Mathlib as the underlying training set (via the Lean Dojo dataset [25]). Our experimental results show that both retrospective rationale generation and expert iteration significantly improve the theorem-proving capabilities of language models in this setting. We describe our setup and findings in detail below.

### 3.1 Experimental Setup

We use *LeanDojo Benchmark 4 v9* as the supervised fine-tuning (SFT) dataset containing $231,240$ data examples. We fine-tune for 1 epoch to obtain the SFT model. For the learning rate, we use a warmup in the first $20\%$ steps from 0 to $2 \times 10^{-5}$, followed by a cosine schedule decaying to zero.

We randomly select $17,256$ different successful proof trajectories from *LeanDojo Benchmark 4 dataset* [25], and use GPT-4-0125 [17] to annotate $52,438$ thoughts from those proof trajectories. We filtered out all proof steps $(s^i, a^i)$ for which $a^i$ contains the newline symbol "\n" before annotating. We perform two iterations of expert iteration, and provide the details in Appendix A.1 due to space.

We evaluate our method on the *MiniF2F* benchmark [28]. We use a similar evaluation setting as previous works [25, 24, 26], but use our sampling method instead of best-first search for the evaluation of our thought-augmented theorem proving model. We choose these settings to resemble the inference budget used in our baselines, which follow previous work [24, 4, 26].

### 3.2 Main Results

Our main results are reported in Table 1. Lean-STaR gives a significant improvement. For instance, with a similar inference budget, Lean-STaR achieves 34.8% versus 30.3% in InternLM2 [26] using best-first search and 30.7% in COPRA [22] using GPT-4. With a larger compute budget, Lean-STaR's performance improves further to 36.1%.

---

[2]For instance, in our experiments we use the best available large language model, GPT-4.

Table 1: **Pass rates on the minif2f-test dataset with Lean.** This table shows the pass rates of previous works and our work. $S$ is the number of tactics attempted at each expanded node (assumed to be 1 in sampling) and $K$ is the total number of search or sampling attempts per problem. In sampling we use temperature 0.7, and in search we use beam search when generating the next tactic. Note that we sample 32 examples twice when $K = 64$ in sampling.

| APPROACH | DECODING | $N$ | $K$ | $S$ | PASS RATE |
|---|---|---|---|---|---|
| GPT-3.5 [1] (FEW-SHOT) | SAMPLING | 50 | 1 | 1 | 2.8% |
| GPT-4 [1] (FEW-SHOT) | SAMPLING | 50 | 1 | 1 | 11.9% |
| TRANSFORMER [19] (W/O RL) | SEARCH | 512 | 1 | 8 | 24.6% |
| LLEMMA-7B [4] (FEW-SHOT) | SEARCH | 50 | 1 | 32 | 26.2% |
| REPROVER [25] | SEARCH | 50 | 1 | 64 | 26.5% |
| TRANSFORMER [19] (W/ RL) | SEARCH | 512 | 1 | 8 | 29.6% |
| INTERNLM2-20B [26] (FEW-SHOT) | SEARCH | 50 | 1 | 32 | 29.5% |
| COPRA (WITH GPT-4) [22] | CUSTOMIZED | - | 100 | 1 | 30.7% |
| INTERNLM2-7B [26] (FEW-SHOT) | SAMPLING | 50 | 32 | 1 | 28.7% |
| INTERNLM2-7B [26] (FEW-SHOT) | SEARCH | 50 | 1 | 32 | 30.3% |
| SFT (INTERNLM2-7B) | SAMPLING | 50 | 32 | 1 | 29.5% |
| **LEAN-COT** (INTERNLM2-7B) | SAMPLING | 50 | 32 | 1 | 32.8% |
| **LEAN-STAR (ITER-1)** (INTERNLM2-7B) | SAMPLING | 50 | 32 | 1 | 34.0% |
| **LEAN-STAR (ITER-2)** (INTERNLM2-7B) | SAMPLING | 50 | 32 | 1 | **34.8**% |
| **LEAN-STAR (ITER-2)** (INTERNLM2-7B) | SAMPLING | 50 | 64 | 1 | **36.1**% |
| INTERNLM2-PLUS-7B [26] (FEW-SHOT) (FROM PAPER) | SEARCH | 1000 | 1 | 32 | 43.4% |
| INTERNLM2-PLUS-7B [26] (FEW-SHOT) (REPRODUCED) | SEARCH | 1000 | 1 | 32 | 42.6% |
| INTERNLM2-PLUS-7B [26] (FEW-SHOT) | SAMPLING | 50 | 32 | 1 | 40.9% |
| SFT (INTERNLM2-PLUS-7B) [26] (FEW-SHOT) | SAMPLING | 50 | 32 | 1 | 41.3% |
| **LEAN-COT** (INTERNLM2-PLUS-7B) | SAMPLING | 50 | 32 | 1 | 43.4% |
| **LEAN-STAR (ITER-1)** (INTERNLM2-7B) | SAMPLING | 50 | 32 | 1 | 45.4% |
| **LEAN-STAR (ITER-1)** (INTERNLM2-PLUS-7B) | SAMPLING | 50 | 64 | 1 | **46.3**% |

**Thought augmentation improves theorem proving.** The first phase of Lean-STaR trains a model to interleave thoughts and tactics, by fine-tuning on a synthesized dataset of thought-augmented examples. The fine-tuned model from this phase, denoted LEAN-COT in Table 1, achieves a pass rate of 32.8%, which is higher than the model prior to this phase, denoted SFT (29.5%). We conclude that the first phase of Lean-STaR can improve the theorem proving ability of a language model, even one that is already specialized for generating tactics in Lean such as the SFT model.

**Bootstrapping improves thought-augmented theorem proving.** The second phase of Lean-STaR consists of generating new thoughts and tactics with the current language model, saving those that result in correct proofs, and training on the union of the initial thought-augmented dataset and the saved examples (i.e., expert iteration [19, 27, 20]). Refer to Appendix A.1 for details.

We perform two iterations of expert iteration, and present the results in Table 1, denoted LEAN-STAR. Each iteration improves the model's theorem proving performance, from 32.8% (the initial model) to 34% (LEAN-STAR after iteration 1) to 34.8% (LEAN-STAR after iteration 2). Furthermore, we find that the model is amenable to further improvement via additional sampling, achieving 36.1% by doubling the sampling budget. We conclude that Lean-STaR's second phase can further improve a model's ability to generate thoughts and tactics that lead to correct proofs. We include three qualitative examples in the Appendix, which show the model interleaving thoughts and proof steps.

### 3.3 Experiments with stronger base model and more data

We also instantiate Lean-STaR using a stronger language model (InternLM2-Math-plus-7b [26]), which was released after the experiment above. Our new results are also reported in Table 1. We can

Table 2: Results for the InternLM2-plus-7b and our Lean-CoT, Lean-STaR, and expert iteration without CoT. We use sampling with $N = 50, K = 32, \& T = 0.7$.

| APPROACH | *Pass@32* OF INTERNLM-BASE | *Pass@32* OF INTERNLM-PLUS |
|---|---|---|
| FEW-SHOT | 28.7% | 40.9% |
| SFT | 29.5%(+0.8%) | 41.3%(+0.4%) |
| LEAN-COT | 32.8%(+**3.3**%) | 43.4%(+**2.1**%) |
| LEAN-STAR | 34.0%(+1.2%) | 45.5%(+**2.1**%) |
| EXPERT ITERATION (SFT) | 30.7%(+1.2%) | 43.0%(+1.7%) |

see that Lean-STaR still gives a significant improvement over the baseline. For instance, Lean-STaR achieves $45.4\%$ versus $39.8\%$ in InternLM-plus using sampling with a similar inference budget and $43.4\%$ using best-first search with more inference budget reported in [26]. This results show that both retrospective rationale generation and expert iteration can improve the theorem-proving capabilities on a stronger base model.

### 3.4 Experiments on expert iteration without CoT

Table 2 shows the result of expert iteration without CoT (i.e., using (state, tactic) pairs only) as well as the result of Lean-CoT and Lean-STaR. Expert iteration alone achieves 43.0%, which is less than Lean-STaR (45.4%) in InternLM-plus and achieves 30.7% verus 39.8% in InternLM-base. This shows that Lean-STaR's performance gains do not only come from the use of expert iteration.

## 4 Conclusion & Limitations

In this paper, we presented Lean-STaR, a novel approach that significantly enhances the theorem-proving capabilities of language models in formal mathematics by integrating Chain-of-Thought (CoT) rationales into each proof step. We further improved this model using expert iteration, fine-tuning it on correct proofs it samples and verifies using the Lean solver. Our contributions include the introduction of the first thought-augmented theorem proving dataset, demonstrating that expert iteration can further improve performance, and achieving much better results on the miniF2F-test benchmark, increasing the pass rate from $30.3\%$ to $36.1\%$. These advancements are not only about improving the accuracy of automated theorem proving, but also offer a scalable and efficient framework for advancing human understanding of mathematics, which may lead to significant impacts in education, scientific discovery, and program verification [8, 12, 21, 3, 10, 16].

The primary limitation of our method is that its performance may be constrained by issues of computational scalability. Both Lean-CoT and Lean-STaR have been fine-tuned on a dataset that is not very large. Additionally, the use of GPT-4 to generate synthetic data may incur a significant cost and possibly introduce biases. Also, expert iteration could face a bottleneck due to CPU and IO limitations, which might slow down the process due to a sluggish speed of Lean ITP.

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

# A   Additional Experiment Setup

## A.1   Lean-STaR Expert Iteration

The second phase of Lean-STaR consists of generating new thoughts and tactics with the current language model, saving those that result in correct proofs, and training on the union of the initial thought-augmented dataset and the saved examples (i.e., expert iteration [19, 27, 20]). We perform two iterations of expert iteration, and provide details on our specific experimental setup below.

In each iteration we use sampling on the *LeanDojo Benchmark 4* dataset, and save the (state, thought, tactic) examples that are part of successful proofs. For each problem, we sample $K = 32$ times in parallel with temperature $T = 1.0$, and limit the number of times a tactic can be generated to a total of $N = 5$ per problem. Also, sampling is limited to 1 minute per problem. In this setup, each problem needs on average about 0.5 A100 minutes. We collect successfully sampled trajectories to produce a "STaR dataset" $D_1$, and up to 3 proof trajectories were collected for each problem. We collected $32, 231$ different (proof state, thoughts, next-tactic) pairs in successful proof trajectories during expert iteration, which takes about 4 days with $8 \times A100$ GPUs. Then, we further fine-tune SFT model for 1 epoch on the combination of GPT-4 annotated reasoning data and expert iteration data $D_T \cup D_1$ to get the Lean-STaR model. We use the same learning rate setup that was used for the SFT model. In the second iteration, we generate a dataset $D_2$ in a similar fashion. Then, we chose to further fine-tune model from iteration 1, $M_1$, on the generated dataset $D_2$ (roughly 19k pairs).

The setup of experiment about InternLM2-plus is slightly different. The details are shown in Section 3.3 and Appendix E.

# B   Statistics for our methods as well as the baselines

Table 3: Statistics for the baselines and our Lean-CoT, Lean-STaR on *MiniF2F* dataset. We use sampling method with hyperparameters $N = 50$ & $K = 32$ & $T = 0.7$.

| APPROACH | # (CONTINUAL) TRAINING DATA | *Pass@32* | |
| --- | --- | --- | --- |
| INTERNLM2-MATH-7B (FEW-SHOT) | - | 28.7% | - |
| SFT | $231, 240$ | 29.5% | +0.8% |
| **LEAN-COT** | $52, 438$ | 32.8% | **+3.3**% |
| **LEAN-STAR (ITER-1)** | $32, 231$ | 34.0% | +1.2% |
| **LEAN-STAR (ITER-2)** | $19, 324$ | **34.8**% | +0.8% |

# C  An Example and Explanation of A Formal Proof in Lean

An example of a formal proof in Lean with its visualization is shown in Figure 2, taken from [13]. In the proof, the tactic `induction k` is is applied to the initial state ($n \leq m \Rightarrow n + k \leq m + k$) and the ITP converts the current state to subgoals `case 0` $\wedge$ `case ih:` $n \leq m \wedge n + k \leq m + k \Rightarrow n + (k+1) \leq m + (k+1)$. The `case 0:` $n \leq m$ is our hypothesis $h_0$ so it can be proven by `case 0:exact` $h_0$ tactic. Then, we rewrite the `case ih` through the `nat.succ_le_succ_iff` which is a theorem in Lean library means $n \leq m \Leftrightarrow n + 1 \leq m + 1$. After tactics `case 0:exact` $h_0$ and `case ih:rw nat.succ_le_succ_iff`, the goal state is converted to $n + k \leq m + k$ which is the hypothesis introduced by induction. Therefore, we can complete this proof using tactic `exact k_ih`.

```
theorem add_le_add_right (m n k : ℕ) (h₀ : n ≤ m)
    : n + k ≤ m + k :=
    induction k with
    | zero =>
        exact h₀
    | succ k ih =>
        rw Nat.succ_le_succ_iff
        exact ih
```

Figure 2: **A example proof and its visualization of** $n \leq m \Rightarrow n + k \leq m + k$ **in Lean, taken from [13].** The `induction` tactic reduces the initial statement to two subgoals. Then tactics `case 0:exact` $h_0$ and `case ih:rw nat.succ_le_succ_iff`, `case ih:exact k_ih` can be applied in turn to complete the proof.

# D Performance Analysis by Types and Difficulties using InternLM2-plus-7b

Table 4 reports the number of problems successfully proved, partitioned by type and difficulty using InternLM2-plus. We see that Lean-CoT improves performance mainly in Number Theory and Lean-STaR improves performance in solving difficult problems on all categories, which is the opposite of the performance of the InternLM2-base.

Table 4: Counts of problems successfully proved in *minif2f-test* benchmark using InternLM2-plus-7b, split by type and difficulty. The methods use sampling with $N = 50, K = 32$.

| TOTAL | | | TEST SET SIZE | INTERNLM2-PLUS-7B | LEAN-COT | LEAN-STAR (ITER-1) |
|---|---|---|---|---|---|---|
| IMO | | | 20 | 0 | 0 | 0 |
| AIME | | | 15 | 3 | 3 | **4** |
| AMC | | | 45 | 9 | 9 | **10** |
| MATH | ALGEBRA | LEVEL 5 | 14 | **6** | 6 | 6 |
| | | LEVEL 4 | 14 | **9** | 9 | 9 |
| | | LEVEL 3 | 14 | 11 | **13** | 13 |
| | | LEVEL 2 | 14 | **11** | 11 | 11 |
| | | LEVEL 1 | 14 | **10** | 10 | 10 |
| | NUMBER THEORY | LEVEL 5 | 16 | 7 | 7 | 7 |
| | | LEVEL 4 | 11 | 6 | **8** | 8 |
| | | LEVEL 3 | 11 | 6 | 7 | **9** |
| | | LEVEL 2 | 11 | 7 | **9** | 9 |
| | | LEVEL 1 | 11 | **10** | 10 | 10 |
| CUSTOM | ALGEBRA | | 18 | **4** | 3 | 4 |
| | NUMBER THEORY | | 8 | 0 | 0 | 0 |
| | INDUCTION | | 8 | **1** | 1 | 1 |

# E   Performance difference of joint training and continue training

As shown in Table 5, the joint training method performs better using InternLM2-base but training method performs much better using InternLM2-plus. It seems that there are no difference between these two methods. Therefore, this performance can be depend on the quantity of data or the model. (We use much more data when using InternLM2-plus and the quantity of "STaR data" is relatively small.)

Table 5: Performance difference of joint training and continue training on Lean-STaR. We use sampling method with hyperparameters $N = 50 \ \& \ K = 32 \ \& \ T = 0.7$. In continue training, we further fine-tune the Lean-CoT model on "STaR data" to get Lean-STaR model and in joint training we fine-tune the SFT model on combination of GPT-4 annotated reasoning data and "STaR data".

| APPROACH | INTERNLM2-BASE-7B | INTERNLM2-PLUS-7B |
|---|---|---|
| **LEAN-COT** | 32.8% | 43.4% |
| **LEAN-STAR (ITER-1)** (JOINT TRAINING) | **34.0%** | 43.9% |
| **LEAN-STAR (ITER-1)** (CONTINUE TRAINING) | 33.2% | **45.5%** |

# F   Retrospective Rationale Generation Prompt of GPT-4

```
Please act as a professional mathematician.
Your goal is to accurately prove a math theorem in Lean4.
You are given the first tactic that should be taken to prove the Given Theorem.

# Example:

Given Theorem:
```lean4
X : Type u_1
Y : Type u_2
inst†¹ : TopologicalSpace X
inst† : TopologicalSpace Y
f : X →. Y
h : PContinuous f
⊢IsOpen (PFun.preimage f Set.univ)
```
Given Tactic:
```lean4
exact h _ isOpen_univ
```
Assistant Output:
### PLANNING OF REASONING
Given the above goal and the given the target tactic `exact h _ isOpen_univ`, I
    should produce a reasoning that only based on the goal, and conclude to the `
    exact h _ isOpen_univ`, but not in the hindsight format.

### REASONING

By directly using the hypothesis `h : PContinuous f`, we state that for any open set
    , its preimage via `f` is also open, and this applies to `Set.univ` as well.

### TACTIC
```

```lean4
exact h _ isOpen_univ
```

To achieve the goal, you have three jobs.
# Write down the planning of reasoning.
# Write down the reasoning that can result in the selection of the Given Tactic.
# Write down the Given Tactic.
You have four principles to do this.
# Craft reasoning that is comprehensive yet concise, contained ideally within two
    sentences.
# Begin reasoning with the objective of the Given Theorem, ensuring logical
    progression, rather than constructing it retrospectively from the Given Tactic.
# Avoid mentioning the Given Tactic and specific statement names from the Given
    Tactic throughout your reasoning.
# Position the conclusion of the tactic after presenting all reasons, ensuring that
    details regarding the Given Tactic are not introduced at the start of your
    reasoning.
Your output should be strictly in the following format and should not contain extra
    content:

### PLANNING OF REASONING

Given the above goal and the given the target tactic <Given Tactic>, I should
    produce a reasoning that only based on the goal, and conclude to the exact <
    Given Tactic>, but not in the hindsight format.

### REASONING

<your reasoning to the Given Tactic>

### TACTIC

<Given Tactic>

# Given Theorem:
```lean4
{theorem["state_before"]}
```

# Given Tactic:
```lean4
{theorem["tactic"]}
```

## G    Examples of generated Lean proofs with thoughts

```
theorem aime_1990_p4 (x : ℝ) (h₀ : 0 < x) (h₁ : x ^ 2 - 10 * x - 29 ≠ 0)
    (h₂ : x ^ 2 - 10 * x - 45 ≠ 0) (h₃ : x ^ 2 - 10 * x - 69 ≠ 0)
    (h₄ : 1 / (x ^ 2 - 10 * x - 29) + 1 / (x ^ 2 - 10 * x - 45) - 2 / (x ^ 2 - 10 *
    x - 69) = 0)
    : x = 13 :=
    norm_num at h₁ h₂ h₃ -- Reasoning: To prove `x = 13`, we examine the conditions
    under which the given equation holds, specifically focusing on the
    non-equality of the expressions involving `x ^ 2 - 10 * x - 29`, `x ^ 2 - 10 *
    x - 45`, and `x ^ 2 - 10 * x - 69`. Simplifying these expressions using
    properties of real numbers and algebra allows us to directly compare them with
    their respective values, ultimately identifying `x = 13` as the unique
    solution that satisfies all constraints.
    field_simp at * -- Reasoning: To prove \(x = 13\), we leverage the given
    equalities involving rational fractions of \(x^2 - 10x - c\) for various
    constants \(c\), where \(0 < x\) and each \(x^2 - 10x - c\) is non-zero. By
    algebraically manipulating these expressions, we aim to derive a relationship
    among \(x\) and these constants, specifically focusing on \(13\), under the
    given non-zero conditions for the quadratic expressions.
    ring_nf at h₄ ⊢ -- Reasoning: To establish the equality `x = 13` under the
    given conditions, we need to manipulate the algebraic expression given by `h₄`
    to reveal its dependence on `x`, recognizing that the presence of `x` in the
    expressions may affect the equation's truth value. The application of
    algebraic transformations on both sides of the equation will facilitate the
    identification of `x`'s value, taking into consideration the inequality
    conditions provided (`h₁`, `h₂`, `h₃`).
    nlinarith -- Reasoning: To prove \(x = 13\), we must manipulate the equation
    given by \(h₄\) to isolate \(x\), using the algebraic properties of equality
    and the assumption \(h₀\) which states \(0 < x\). This requires careful
    rearrangement and application of algebraic identities to simplify and
    rearrange terms, facilitating the direct application of the properties of real
    numbers to reach the desired conclusion.

theorem amc12a_2002_p6 (n : ℕ) (h₀ : 0 < n)
    : ∃ m, m > n ∧ ∃ p, m * p ≤ m + p :=
    use n + 1, by linarith only [h₀] -- Reasoning: To demonstrate the existence of
    such an `m` and `p` that satisfy the given inequalities, we leverage the
    property of natural numbers that ensures an increase in `n` from `0` leads to
    the possibility of finding such `m` and `p`. This approach is supported by the
    fundamental property of natural numbers and their arithmetic operations,
    suggesting that for a given `n` greater than `0`, there exists a successor `m`
    and a number `p` that satisfy the conditions outlined.
    exact ⟨0, by simp ⟩ -- Reasoning: To demonstrate that there exists an \(n \in ℕ
    \) such that \((n + 1) * p \leq n + 1 + p\) holds for some \(p \in ℕ\), we
    recognize that setting \(p = 0\) satisfies the inequality by the properties of
    zero in arithmetic. This is because \(n + 1 > 0\) by assumption, ensuring the
    left-hand side is non-negative and the right-hand side simplifies to \(n +
    1\), making the inequality trivially true.
```

theorem aime_1983_p2 (x p : ℝ) (f : ℝ → ℝ) (h₀ : 0 < p ∧ p < 15) (h₁ : p ≤ x ∧ x ≤ 15)
    (h₂ : f x = abs (x - p) + abs (x - 15) + abs (x - p - 15))
    : 15 ≤ f x :=
  cases' h₁ with h₁ h₂ -- *Reasoning: Given a function \(f\) defined over the real numbers, where \(f(x)\) equals the sum of the absolute values of the differences \(x - p\), \(x - 15\), and \(x - p - 15\), along with conditions on \(p\) and \(x\) that ensure \(x\) lies between \(p\) and \(15\) and \(p\) is positive, we aim to show that \(f(x) \geq 15\). To demonstrate this, we consider all possible cases for the position of \(x\) relative to \(p\) and \(15\), specifically at \(p\) itself, between \(p\) and \(15\), and at \(15\). By evaluating \(f(x)\) for each case, we can establish a lower bound of \(f(x)\) that is greater than or equal to \(15\).*
  simp [h₀, abs_of_nonneg (sub_nonneg.2 h₁), abs_of_nonneg (sub_nonneg.2 h₂), *]
  -- *Reasoning: The objective is to show that \(f(x)\) is at least 15 for \(x\) within the bounds \(p \le x \le 15\), where \(p\) is positive but less than 15. The provided expression for \(f(x)\) suggests using properties of absolute values and inequalities to simplify and analyze \(f(x)\). By applying simplification rules for absolute values and inequalities, along with the given bounds, we aim to demonstrate the desired inequality \(f(x) \ge 15\).*
  simp only [abs_of_nonneg] at * -- *Reasoning: The objective is to establish an inequality involving an expression with absolute values and a sum, where all terms are functions of \(x\). By recognizing that the absolute value functions are always non-negative, we can simplify the given expression by removing the absolute value symbols, thus focusing on the arithmetic manipulation of the terms to demonstrate the inequality.*
  simp only [abs_of_nonneg, sub_add_eq_add_sub] at * -- *Reasoning: The objective is to demonstrate that the expression `15 ≤ x - p + |x - 15| + |x - p - 15|` holds given the conditions on `p`, `x`, and the function `f`. Since the expression in question involves absolute values which inherently involve non-negativity, manipulating these expressions to highlight their non-negativity or equality is a logical initial step. This involves recognizing that the absolute value of a difference can be rewritten as the difference of absolute values, which simplifies the manipulation of the expression to prove the inequality.*
  congr 1 -- *Reasoning: To establish the inequality `15 ≤ x + |x - 15| + |x - p - 15| - p`, we recognize that the left-hand side of the inequality involves a mix of additions and subtractions of real numbers. By introducing a congruence transformation that simplifies operations involving absolute values, we aim to align both sides of the inequality in a form that directly reflects the properties of real numbers under these operations.*
  simp only [abs_of_nonpos (sub_nonpos_of_le h₂), abs_of_nonneg (sub_nonneg_of_le h₁)] -- *Reasoning: The objective is to prove an inequality involving absolute values and subtractions that aligns the given conditions and constraints. Simplification using the properties of absolute values and the provided bounds can directly lead to the desired conclusion by expressing the absolute values in terms of their underlying expressions and applying the given constraints.*
  simp [abs_of_pos, *] -- *Reasoning: To prove the given inequality involving absolute values, simplification using algebraic properties and given hypotheses will directly lead us to the desired inequality by recognizing the algebraic manipulation and application of the known bounds `h₁` and `h₂`, as well as the non-negativity of `p`.*
  rw [abs_of_nonpos] -- *Reasoning: The objective is to demonstrate that `15 ≤ 15 + |x - p - 15| - p` under the given constraints on `p` and `x`. Given `p < 15` and `p ≤ x ≤ 15`, it's apparent that `p - 15` is non-positive. This observation allows us to apply the property that the absolute value of a non-positive number is equal to its negative value, thereby simplifying the expression on the right-hand side of the inequality.*
  all_goals linarith -- *Reasoning: The goal involves demonstrating a relationship between expressions involving inequalities and subtraction. The simplification of these expressions into a form that directly compares their numerical values can lead to a straightforward application of known inequalities and algebraic properties, demonstrating the inequality's validity under the given conditions.*

