# OpenReview forum: "Lean-STaR: Learning to Interleave Thinking and Proving"
_NeurIPS.cc/2024/Workshop/MATH-AI — MATH-AI 24_

### Official Review · Reviewer_ZLMT · 2024-10-02
**Bootstrapping thought augmentation for theorem proving**

**Rating:** 6
**Confidence:** 5

**Review:**

This paper integrates self-taught reasoning into the theorem proving task. The observation that informal thought processes should be able to help models with formal theorem proving sounds right, and the method is well executed. The data curation and the training look sound, and the evaluation is comprehensive and well-compared. I think it's okay to apply a method to a new field as long as it works well there, so I am leaning towards accepting this work, also given that this work has been quite solid and complete. By the way, in the last "conclusion & limitations" section there seems to be no limitations discussed, just conclusion alone?

---

### Official Review · Reviewer_BLki · 2024-10-08
**The paper presents Lean-STaR, a framework designed to enhance the theorem-proving capabilities of language models within the Lean interactive theorem prover. By interleaving informal "thoughts" with formal proof steps, the approach mirrors human reasoning processes, addressing a gap in traditional language model-based theorem proving that typically relies solely on formal proof data.**

**Rating:** 7
**Confidence:** 3

**Review:**

Quality:
The paper provides a well-structured and methodologically sound study on enhancing theorem-proving capabilities of language models within the Lean theorem-proving environment. The authors introduce Lean-STaR, a framework that interleaves informal "thoughts" with formal proof steps, effectively mimicking human reasoning processes. The experimental design seems robust, involving the creation of a thought-augmented dataset and the application of expert iteration for model fine-tuning. The results are compelling, showing  improvements over baseline models on the miniF2F-test benchmark.

Clarity:
Overall, the paper is clearly written and logically organized. The introduction effectively sets the context and motivation for the work, highlighting the gap in traditional language model-based theorem proving that Lean-STaR aims to fill. The methodology section provides a detailed explanation of the two-phase training process:
- Thought-Augmented Tactic Prediction: Introducing a hidden "thought" variable prior to each tactic, enhancing the model's ability to predict the next tactic based on both the current state si and the thought si
- Bootstrapping via Expert Iteration: Further refining the model by generating new thought-tactic pairs from successful proofs and fine-tuning the model iteratively.

Originality:
The paper seems to make a novel contribution by integrating informal reasoning steps into formal theorem proving within an interactive theorem prover. While previous works have focused on formal proof data alone, this approach acknowledges the value of the informal thought processes that humans naturally engage in when constructing proofs. The creation of the first thought-augmented dataset for theorem proving also seems to be an original and significant contribution.

Significance:
The work seems to have substantial significance in the fields of automated theorem proving and formal methods. By demonstrating that interleaving informal thoughts with formal proof steps can enhance theorem-proving capabilities, the paper might open new avenues for research in combining symbolic and neural methods. The improvements achieved on the miniF2F-test benchmark suggest practical benefits for advancing automated reasoning systems. Additionally, the approach has broader implications for education, scientific discovery, and program verification, potentially impacting how formal reasoning tasks are approached in various domains.

Pros:
- The approach bridges the gap between human-like informal reasoning and formal proof generation, introducing a new method that enhances theorem-proving capabilities.
- Lean-STaR achieves higher pass rates on the miniF2F-test benchmark compared to baseline models, demonstrating the effectiveness of the method.
- The combination of thought augmentation and expert iteration provides a comprehensive training process that incrementally improves the model's performance.
- Generating approximately 100,000 thought-augmented examples contributes valuable resources to the research community.
- The inclusion of examples showing how thoughts are interleaved with proof steps offers transparency into the model's reasoning process.

Cons:
- The experiments are primarily conducted within the Lean environment on the miniF2F-test benchmark, which may limit the generalizability of the results.
- A lack of detailed information on training hyperparameters and settings may hinder reproducibility and replication efforts.
- The paper could benefit from more in-depth ablation studies to isolate and quantify the contributions of individual components, such as thought augmentation and expert iteration.

---

### Official Review · Reviewer_msBK · 2024-10-08
**Review of Lean-STaR: Learning to Interleave Thinking and Proving**

**Rating:** 6
**Confidence:** 3

**Review:**

Summary:
This paper proposes Lean-STaR, a method for improving theorem proving by integrating informal "thoughts" with formal tactics. The approach builds on the STaR framework which uses expert iteration to fine-tune the model on correct proofs with (state, thought, tactic) triplets. The results show improvements in pass rates on the miniF2F benchmark.

Strengths:
- The combination of informal thoughts and formal tactics is a promising direction for theorem proving, also a good extension to the STaR framework.
- The expert iteration phase provides consistent performance gains, with clear improvements across iterations (in my experience, expert iteration can sometimes lead to performance regressions after several iterations, possibly due to the loss of plasticity or collapse; it would be interesting for the authors to explore the effects of continuing self-training for more iterations and see if these potential issues emerge, as this could provide new insights into model stability).
- The proposed method has the potential to inspire further research in thought-augmented reasoning for formal verification tasks.

Cons / Comments:
- Comparison Table: There is a major concern with the fairness of the comparison table. Results reported from different papers may have been obtained under different hardware/software settings. It is crucial for the authors to clearly indicate which results were reproduced under consistent conditions and which were directly taken from other works. Without this clarification, the comparison may mislead readers unfamiliar with the literature, as it is not an apple-to-apple comparison in several respects.
- More ablation studies would strengthen the claims, especially comparing models trained without thought augmentation (i.e., SFT-only expert iteration).
- Adding experiments on additional models (e.g., mistral, gemma, llemma) and evaluation datasets (e.g., leandojo) would improve the generalizability of the results.

Overall, I believe the contributions of this work is sufficient as a workshop paper. It would be exciting to see further explorations in the domain of thought-augmented theorem proving. The current results are promising, and extending the approach with investigations in the exploration v.s. exploitation of thought generation in theorem proving could have a broader impact.

---

### Official Review · Reviewer_wGTV · 2024-10-08
**Reivew of "Lean-STaR: Learning to Interleave Thinking and Proving"**

**Rating:** 6
**Confidence:** 3

**Review:**

# Summary
"Lean-STaR: Learning to Interleave Thinking and Proving" explores the hypothesis that informal information, which is typically absent in formal proofs, can significantly aid in the process of learning to prove theorems. The paper makes several noteworthy contributions to the field:
1. Introduction of a Thought-Augmented Theorem Proving Dataset: This dataset is the first of its kind and is designed to incorporate 'thoughts' or informal reasoning steps that precede formal proofs.
2. Demonstration of Expert Iteration: The paper shows that expert iteration can enhance performance in theorem proving tasks.
3. Improved Performance on the miniF2F-test Benchmark: By implementing the proposed methods, the paper reports an increase in the pass rate from 30.3% to 36.1% on this benchmark, indicating a significant improvement over previous methods.
# Evaluation
The concept of interleaving thinking with proving is intriguing and aligns well with the intuitive processes humans often use in problem-solving. The introduction of a thought-augmented dataset is particularly innovative and could pave the way for new research directions in automated theorem proving.
However, the paper could benefit from clearer examples and comparisons to traditional methods. The examples provided, particularly in Section 1 Introduction, could be more illustrative of the advantages of the proposed multi-step method over conventional approaches. The necessity and nature of the intermediate 'thought' steps are not immediately apparent, which may leave readers questioning the practical utility and efficiency of the proposed method.
A specific concern arises with the example used in the introduction to find the value of 'b'. The method employed appears unconventional and not generally applicable to similar types of problems. More critically, the example concludes without a formal verification of the result ('b=248'), which undermines the effectiveness of the Lean-STaR approach in this instance. This could potentially mislead readers about the capabilities and reliability of the proposed method.
# Recommendation
While the paper presents a novel approach and makes significant contributions to the field of automated theorem proving, I recommend a revision to address the clarity and applicability of the examples provided. Enhancing the comparative analysis with traditional methods and ensuring that all examples robustly demonstrate the efficacy of the approach will make the contributions more compelling and the paper stronger overall.
Given these adjustments, the paper has the potential to make a substantial impact on the field, and I would lean towards recommending it for marginal acceptance.

---

### Official Review · Reviewer_YsMe · 2024-10-08
**A novel method with convincing results**

**Rating:** 8
**Confidence:** 4

**Review:**

## Summary
The paper suggests to use the Self-Taught Reasoner (STaR) framework for training chains-of-thought (CoTs) in a batch reinforcement learning fashion with rejection sampling. Pretrained LLMs are used to post-hoc rationalize Lean proof tactics, yielding a CoT-augmented dataset of Lean proofs. After this, expert iteration (= batch reinforcement learning) is applied to generate additional training data from problem statements and the model's successful (CoT-augmented) proof trajectories

The paper is well-written and tackles the important question of reinforcement learning for theorem proving with very good results on minif2f-test (Lean4).

## Weaknesses
- While the results are convincing, it is unclear how much of the improved performance should be attributed to the supervised finetuning and the expert iteration method, compared to the CoT-augmentation technique itself. Judging from the provided proofs in the appendix, a typical non-thought augmented language model should also have come up with such proofs since they (as many language model written Lean4 proofs on minif2f) rely heavily on Lean's automation tactics. Sequence distillation from GPT4 may have contributed to the +3.3 points observed for thought augmentation (which I consider valid to use but the question remains whether sequence-distilling CoTs is the most efficient method to put pretrained LLMs to use in automated theorem proving). *Beyond the scope of this workshop paper,* I would be interested in more ablations regarding the contribution of sequence distillation, CoT and expert iteration to the method's results, respectively, and in comparisons to related methods of increasing inference time token/compute budget and using large models.
- InternLM is possibly contaminated for minif2f since minif2f (Lean3) repositories with proofs were likely included in its training data.

---

### Decision · Program_Chairs · 2024-10-08

Accept